# Autotransplantation of the Third Molar: A Therapeutic Alternative to the Rehabilitation of a Missing Tooth: A Scoping Review

**DOI:** 10.3390/bioengineering8090120

**Published:** 2021-09-02

**Authors:** Mario Dioguardi, Cristian Quarta, Diego Sovereto, Giuseppe Troiano, Michele Melillo, Michele Di Cosola, Angela Pia Cazzolla, Luigi Laino, Lorenzo Lo Muzio

**Affiliations:** 1Department of Clinical and Experimental Medicine, University of Foggia, Via Rovelli 50, 71122 Foggia, Italy; cristian_quarta.549474@unifg.it (C.Q.); diego_sovereto.546709@unifg.it (D.S.); giuseppe.troiano@unifg.it (G.T.); michele.melillo@hotmail.it (M.M.); dott.dicosola@gmail.com (M.D.C.); elicio@inwind.it (A.P.C.); lorenzo.lomuzio@unifg.it (L.L.M.); 2Multidisciplinary Department of Medical-Surgical and Odontostomatological Specialties, University of Campania “Luigi Vanvitelli”, 80121 Naples, Italy; luigi.laino@unicampania.it

**Keywords:** tooth autotransplantation, third molar, oral surgery

## Abstract

Introduction: Tooth autotransplantation is the repositioning of an erupted, partially erupted, or non-erupted autologous tooth from one site to another within the same individual. Several factors influence the success rate of the autotransplant, such as the stage of root development, the morphology of the tooth, the surgical procedure selected, the extraoral time, the shape of the recipient socket, the vascularity of the recipient bed, and the vitality of the cells of the periodontal ligament. The aim of this scoping review was to provide the most up-to-date information and data on the clinical principles of the third-molar autograft and thus provide clinical considerations for its success. Materials and methods: This review was conducted based on PRISMA-ScR (Preferred Reporting Items for Systematic reviews and Meta-Analyses extension for Scoping Reviews). The research was conducted by searching for keywords in three databases—PubMed, Scopus and Google Scholar—by two independent reviewers following the PRISMA protocol, from which 599 records were identified. Conclusions: Third-molar autotransplantation is a valid solution to replace missing teeth. The key to the success of this technique is the surgical procedure, which must be as atraumatic as possible to preserve the periodontal ligament of the tooth to be transplanted. The success rate is also linked to the stage of development of the root, with a worse prognosis in the case of a complete root.

## 1. Introduction

Tooth autotransplantation is the repositioning of an erupted, partially erupted, or non-erupted autologous tooth from one site to another within the same individual. It was first reported in 1950 as an alternative to replacing a non-restorable tooth [1,2]. Autotransplantation is a means of replacing a tooth that is missing or requires extraction due to tooth decay, periodontal disease, or another reason [3]. The procedure may include extraction of the tooth from the recipient site, preparation of the recipient socket, atraumatic extraction of the donor tooth, minimal extraoral time, positioning and stabilization of the donor tooth, and any occlusal adjustments [4].

Several factors influence the success rate of autotransplantation, such as the stage of root development, the morphology of the tooth, the surgical procedure selected, the extraoral time, the shape of the recipient socket, the vascularity of the recipient bed, and the vitality of the cells of the periodontal ligament [5]. Avulsed teeth recover optimal function and aesthetics after replanting under ideal conditions.

Although a dental implant offers the most desirable treatment option for a missing tooth, it is contraindicated in children and adolescents due to the continued growth of the alveolar process, which poses the risk of severe imposition [6]; therefore, autotransplantation offers an alternative in these cases. Generally, the most commonly transplanted teeth are premolars, canines, incisors, and third molars [7].

From a clinical perspective, the transplantation of a third molar to replace an untreatable tooth is a valid alternative to prosthetic or implant rehabilitation [8]; moreover, the advantages of an autotransplanted tooth over a dental implant include the maintenance of proprioception, possible orthodontic movements, relatively low cost, and pulp regeneration in immature teeth [4,9].

The outcome of the autotransplanted tooth can be defined as follows:Success: no evidence of root resorption or ankylosis, inflammation, immobility, or periodontal pockets, and no pain in function;Survival: no pain, no inflammation but with root resorption or ankylosis;Failure or pathology: more than 3 mm of pocket from the end of the first year of transplantation, pain in function, abnormal mobility, infection at the recipient site.

To ensure success, the autotransplant requires meticulous selection of candidates and procedural planning. Selection criteria include factors such as general health, psychological behavior, willingness to undergo the procedure, and, most importantly, oral hygiene status [10]. Both the donor tooth and the recipient site should be examined with the utmost care to ensure compatibility and fit [11].

Previous literature reviews from the past decade have investigated various functional, surgical, orthodontic, and biological implications of dental autografts. Plotino et al., in a recent narrative review, identified the success of surgical and endodontic autotransplantation procedures as the absence of dental resorption and ankylosis. The authors also included the absence of apical periodontitis, which is not a priority in many observational studies on autotransplantation [12].

In a systematic review, Lacerda-Santos investigated the aspects related to orthodontic treatments to which the transplanted teeth are subjected, concluding that in the self-transplanted teeth, there is an increase in root resorption influenced by orthodontics, but without affecting the general long-term clinical result. The authors in that study state that “bone and periodontal tissue do not appear to be significantly affected by orthodontics” [13]. In agreement with this review, Hariri notes, regarding autotransplants in orthodontics, that transplantation has other advantages over tooth replacement, the most important of which is the potential for bone induction and the re-establishment of a normal alveolar process. Even if the transplant fails later, there is an intact recipient area that could be used for an implant [14].

In a review of observational studies conducted on autotransplantation of third molars, Armstrong et al. identified the favorable prognostic factors that are related to the patient (patient’s health conditions and motivations), the tooth (absence of root anomalies), and surgical techniques (the most atraumatic possible) [15].

In four systematic reviews of the literature, Martin et al. 2018 identified a success and survival rate of >81% (with five-year survival rates reported to be as high as 80.5%) [16]. For Rholf (immature apex), this rate increased to around 95% (from 1 to 5 years) and 90% at the 10-year follow-up [17].

These previous reviews highlight many aspects inherent to autotransplants, focusing on the surgical protocols adopted and the influence of orthodontic treatments, and indicating the different success rates.

In contrast, in the current review, we aim to provide the most up-to-date information and data on the clinical principles of autotransplantation of the third molar, and thus, provide clinical considerations for its success.

## 2. Materials and Methods

This review was conducted based on PRISMA-ScR (Preferred Reporting Items for Systematic reviews and Meta-Analyses extension for Scoping Reviews) [18]. The results, the methods of data extraction, and the methods for their quantitative and qualitative synthesis were previously agreed upon by three independent reviewers.

After an initial screening phase, eligible works were included in a qualitative analysis and conclusions were evaluated to determine the most up-to-date information relating to third-molar autotransplantation.

The exclusion criteria were: non-English articles, systematic reviews and reviews, in vitro studies, studies on animal models, and studies conducted on transplants of teeth other than third molars. No filters were applied for year of publication.

The studies considered for inclusion in the qualitative analysis were: clinical studies, case series, and case reports relating to dental autotransplants of third molars published in English and those that reported the most current and significant data and information regarding prognosis, management, surgical method, and survival of autotransplants.

The five researched outcomes were:I.Primary outcome: Evaluation of the maturation stage of the third molar to be transplanted (Section 4.1);II.Secondary outcome: Methodology of atraumatic extraction of the third molars to be transplanted and periodontal prognostic factors (Section 4.2 and Section 4.3);III.Tertiary outcome: Preparation techniques of the receiving site (Section 4.4);IV.Quaternary outcome: Positioning and stabilization techniques of the transplanted tooth (Section 4.5);V.Quinary outcome: Endodontic treatment of the transplanted tooth (Section 4.6).

Studies were identified using electronic databases and examining the references in the retrieved articles. The bibliographic research was conducted using the PubMed and Scopus search engines. The electronic database search was conducted between 2 January 2021 and 1 February 2021, and the latest search for a partial literature update was conducted on 6 March 2021.

The following search terms were entered in PubMed and Scopus: “third molar autotransplantation” and “tooth autotransplantation”. Filters for systematic reviews, reviews, and clinical studies were applied to search for terms to identify previous systematic reviews, and thereby, not replicate results and hypotheses already taken into consideration. After identifying the records, the overlaps were removed using Endnote software.

This research concerns the subsequent screening of the results obtained from the search, which was carried out by two independent reviewers; uncertain positions were discussed with a third reviewer. The screening included the analysis of the title and the abstract to eliminate the articles not relevant to the topics of the review. The potentially admissible articles were finally subjected to a full-text analysis to verify their use for the purpose of qualitative analysis. Disagreements were resolved by a third reviewer and a fourth reviewer oversaw the entire study.

The two independent reviewers were M.D. and C.Q., the third reviewer was G.T., and the fourth reviewer, who oversaw the project, was L.L.M., all of whom are dentists of the Department of Clinical and Experimental Medicine of the University of Foggia (Italy).

## 3. Results

From the PubMed and Scopus searches, 532 records were identified; from Google Scholar, 67 records were identified. Using EndNote software, overlaps were removed, resulting in 289 records. Following the initial application of the eligibility criteria (non-English abstracts and issues inconsistent with dental self-transplants), 249 articles were obtained. A total of 180 articles were initially deemed eligible for review, and 78 articles were finally included in the review as agreed upon by the two independent reviewers.

The 31 included studies for the qualitative analysis were:I.Primary outcome: Schliephake and NeuKam 1990 [19], Moorrees et al. 1963 [20], Atala-Acevedo et al. 2017 [21], Lundberg and Isaksson 1996 [22], Lucas-Taulé et al. 2021 [23], Rey Lescure et al. 2021 [24], Tang et al. 2017 [25], and Jang et al. 2013 [9].II.Secondary outcome: Lucas-Taulé et al. 2021 [23], Nagata et al. 2016 [26], Kristerson et al. 1991 [27], Bauss et al. 2004 [28], Sugai et al. 2010 [3], Jang et al. 2016 [29], Aoyama et al. 2012 [30], Koszowski et al. 2013 [31], He et al. 2018 [32], and Shahbazian et al. 2010 [33].III.Tertiary outcome: Arbel et al. 2019 [34], Mena-Álvarez et al. 2020 [35], Bauss et al. 2004 [36], Devi et al. 2014 [37], and Alkofahi et al. 2020 [38].IV.Quaternary outcome: Motegi et al. 2009 [39], Bauss et al. 2005 [40], and Gault and Warocquier-Clerout 2002 [41].V.Quinary outcome: Dharmani et al. 2016 [42], Boschini et al. 2020 [43], Lin et al. 2020 [44], Mejàre et al. 2004 [45], and Kumar et al. 2013 [46].

All selection and screening procedures are described in the flow chart shown in Figure 1, and details regarding the keywords used in the different databases are shown in Table 1.

## 4. Discussion

### 4.1. Surgical Procedure

The operative protocol first of all provides for local anesthesia at the level of the inferior alveolar nerve (in the mandible) or of the posterior superior alveolar nerve (in the maxilla). Then, a triangular flap is drawn for access to the surgical site, in case the third molar is included in the bone [10], or an intracrevicular incision is made, in order to preserve the periodontal ligament, followed by dislocation of the tooth, if the tooth is present in the oral cavity [47], thus proceeding with the extraction of the third molar in an atraumatic way; the extracted tooth is then stored in Hank’s balanced saline solution or [47], alternatively, in pasteurized milk [48]. Then, the recipient site is prepared by means of round surgical burs at low speed and cooled with saline solution [4]. The donor tooth is then inserted and kept out of occlusal contact to avoid destabilizing occlusal forces, preferring a more flexible splint than a rigid one to facilitate pulp revascularization [49]. The surrounding soft tissue is then repositioned and sutured with resorbable or non-resorbable wires.

The patient should be educated about postoperative management through oral hygiene indications, which includes rinses with 0.12% chlorexedine gluconate, liquid diet and soft foods, antibiotic therapy and the use of anti-inflammatory drugs as needed [11].

After one week, it is possible to remove the suturing threads, while the splint is removed after 2–4 weeks [10,50]. Endodontic treatment is performed after 2–3 weeks in the case of teeth with complete root formation to prevent the spread of infection of the pulp from the periapical area and the consequent inflammatory resorption of the root [3] (Figure 2). A re-evaluation should be done at 1, 3, 6, 9, and 12 months, to clinically evaluate mobility, sensitivity to percussion, and probing depth, and radiographically evaluate the presence of signs of inflammation, bone resorption, or disappearance of the periodontal space [50].

### 4.2. Stage of Maturation of the Third Molar

Teeth transplanted with an incomplete root have a pulp healing rate of 96%, compared with 15% for transplanted teeth with complete root formation [51].

Dental elements subjected to autotransplantation procedures undergo a series of complications, the most common of which are root related, such as root ankylosis and resorption, whereas others affect the pulp tissue, such as necrosis [52]. These complications appear to be related to the stage of root maturation and its ability to revascularize; a root with an immature apex and a degree of formation of 75% would appear to have a greater capacity for revascularization and apexification as also recently reported in a study by Erdem and Gümüşer [9,53,54]; some authors prefer radiographic evidence that indicates the root has developed at least 2–3 mm, whereas others claim a root development of at least 3–5 mm [55,56]. The regeneration capacity of the pulp vascular tissue, due to the capillaries coming from the root apex that is not yet fully formed, preserves the vital tooth after the self-transplant [57].

As a result of immature third molars having a rich supply of blood and stem cells, root development following transplantation depends on the preserved activity of Hertwig’s epithelial sheath; in fact, its presence not only translates into root development, but also affects periodontal healing [58,59]. In this regard, a review of the literature conducted by Gugliandolo et al. highlighted the possible use in tissue bioengineering of oral mesenchymal stem cells in particular deriving from dental tissues, such as dental pulp stem cells and stem cells from the apical papilla (periodontal ligament stem cells, gingival-derived steam cells, dental follicle stem cells, and tooth germ stem cells) [60]. It is, therefore, recommended that immature teeth be used from late stage 2 (half root formation) to stage 4 (three-quarters to less than complete root) of root development [9]. In addition, for the correct growth of the alveolar ridge, it is essential to preserve the vitality of the tooth whenever possible to prevent ankylosis or root resorption [9,20,61] (Figure 3) [62].

To confirm the previous statements, a histometric study was conducted by Shlephake and NeuKam which related the periodontal damage caused during extraction to the state of maturation of the root, reporting a greater amount of tissue damage in an advanced state of the root ripening [19].

A systematic review conducted by Atala-Acevedo in 2016 confirmed the effect of the maturation stage of the open apex root on the success of autografts. The reported success rate for molars was 98.21% and the mean follow-up period was 6 years and 3 months, with a higher success rate for premolars than molars [21].

The results of this review of meta-analyses were partially confirmed by a study by Tang et al. 2017 [25] and many case reports (Jang et al. 2013 [9], Rey Lescure 2021 [24]). A subsequent study by Lucas-Taulé et al. 2021, which reported rates of success in both open and closed apexes of 92–97%, concluded that the state of maturation of the root does not correlate with the success rate [23].

### 4.3. Atraumatic Extraction of Third Molars

A consensus exists among the authors that well-performed surgical procedures are important to ensure minimal trauma to the root surface [63,64,65]. Care must be taken not to damage the periodontal ligament of the donor tooth; before dislocation, an intracrevicular incision is made to preserve the periodontal ligament and the tooth is extracted as slowly and atraumatically as possible. The tooth is then preserved during the preparation of the recipient site in Hank’s balanced saline solution [47]. The temporary storage medium is a factor that can influence the prognosis, with Hank’s balanced saline solution being the most suitable for maintaining the vitality of the cells of the periodontal ligament, followed by pasteurized milk, which is considered a valid alternative [48]. The advantages of milk include its physiologically compatible pH, osmolality with the cells attached to the root surface, and the presence of nutrients and growth factors [10].

The results of a clinical study conducted on 25 transplanted third molars suggest the use of drills under irrigation with normal saline solution during the preparation of the donor site [63]. To reduce the trauma, the bur must be round and operated at low speed [8]. Piezosurgery using tips with certain vibration frequencies can be used for the autotransplantation of non-erupted third molars to facilitate their removal from the bone, with few lesions on the periodontal fibers or the follicular sac, and to reduce the occurrence of ankylosis or root resorption [31].

The longer the time interval between extraction and transplant, the more negative the prognosis. The prognosis of autotransplanted teeth is better when this period is shorter [4,30,31].

An adequate period of extraoral time is important to preserve the vitality of the periodontal ligament cells, which should remain viable for 18 min. After this time, the cells become hypoxic and may subsequently go into necrosis, resulting in inflammatory root resorption [66].

The preparation of the recipient site requires a considerable amount of time because several tests of the donor tooth are required for it to have adequate positioning; thus, there is a risk that the tooth will remain extraoral for a long period. To significantly reduce this time (more than 30 s), the use of a printed stereolithographic replica has been suggested to provide the time necessary for the preparation, thus avoiding a long period in which the tooth to be transplanted is outside of the oral cavity [50]. This also reduces the number of attempts to position the tooth in the socket, which must be as low as possible to avoid damage to the periodontal ligament. It is now possible to perform a CBCT scan before surgery to obtain the necessary data on the donor tooth to design and manufacture its replica [32]. This process has an accuracy of 0.25 mm, which is satisfactory in most cases [33,67].

### 4.4. Importance of the Periodontal Ligament in Transplant Success

The periodontal ligament is one of the most important factors for the success of the transplanted tooth [68]. Reattachment takes place within about two weeks after the autotransplant between the connective tissues of the periodontal ligament of the root of the donor tooth and the wall of the recipient alveolus. The periodontal ligament has cells that genetically have the ability to differentiate into fibroblasts, osteoblasts, and cementoblasts. In particular, they play a fundamental role in tissue regeneration periodontal ligament stem cells that are collected by scraping the alveolar ridge and the horizontal fibers of the ligamentous periodontal tissue of dental elements with healthy tissue, as reported in a recent review of the literature conducted by Trubiani et al. [69,70], which was partially confirmed by a subsequent study by Marconi et al. [71]. In the ideal situation, it is hoped that the periodontal ligament cells on the external apical surface of the tooth will differentiate into cementoblasts and stimulate the deposition of dentin, whereas the cells facing the surface of the bone socket wall differentiate into osteoblasts, thus stimulating the bone formation. Root surface healing depends on the extent of damage to the root surface. Healing can be achieved by cementitious healing for small damaged periodontal surfaces; however, when the extent of damage is large, resorption of the root surface and replacement of dentin with bone occurs, which leads to the loss of the root of the tooth [4].

In 1982, Lindskog and Blomlöf examined the periodontal healing process. They found that the periodontal ligament is sensitive to changes in osmotic potential and acidity [72], and fibroblasts die in the case of prolonged exposure to an extraoral environment. Additionally, inflammatory root resorption and ankylosis can occur if the drying time exceeds 30 and 60 min, respectively [68].

Some authors have reported that the regenerative potential of periodontal ligament cells is reduced with age, which could, therefore, interfere with the normal adaptation of the donor tooth in the recipient site [73].

The careful and accurate extraction of the donor tooth is highly important for the preservation of the periodontal ligament [74]. It has been shown that bone regeneration can be induced in the recipient site after transplantation when the periodontal ligament cells of the donor tooth are preserved [75,76].

The role of the periodontal ligament and its integrity has been proven to be an important prognostic factor, as demonstrated in a recent study by Lucas-Taulé et al. 2021, which monitored periodontal indices in transplanted third molars: probing pocket depth (PPD), gingival recession (REC), and clinical attachment level (CAL) [23]. Lucas-Taulé et al. confirmed the results of a study of 18 patients conducted by Kristerson et al. 1991, which showed that the autotransplantation of third molars to replace molars lost due to periodontal disease represents a valid treatment [33].

Sugai et al. suggest that periodontal negative prognostic factors that can influence success may be probing pocket depth of the donor tooth greater than 4 mm, age greater than 40 years, and endodontic treatment [33]. These data agreed with Aoyama et al. 2012, who also reported the rotation of the donor tooth as a negative prognostic factor [30].

The periodontal integrity of the tooth to be transplanted has been found to play a key role in its survival. Orthodontic treatment does not appear to negatively influence the periodontal status of the third molar, as reported by Bauss et al. 2004, who examined 91 third molars that were transplanted and subjected to derotation and orthodontic extrusion [28].

In addition, the presence of cement tears also represents an unfavorable prognostic factor in the survival of dental elements subjected to autotransplantation due to the onset of resorption phenomena in this regard. Nagata et al. 2016 described a successful case of a molar with a cement tear surviving for 15 years [26].

#### Materials Used as Scaffolds for Tissue and Periodontal Regeneration of Transplanted Teeth

One of the main reasons for the loss of the autograft tooth is the damage caused to the periodontal apparatus during surgical maneuvers, which subsequently leads to root resorption. Some studies report that the third molar could potentially be used as a scaffold for tissue regeneration of the periodontal ligament; in fact, Mino et al. evaluated the adhesion of HPDL cells (human periodontal ligament) on the root surfaces in order to regenerate the periodontal membrane on a sterile root surface in vitro before transplantation [77], obtaining the adhesion of the periodontal ligament cells on the tooth surface sterilized. A bioabsorbable polymeric material 3D printed after design and transplanted with HPDL cells was also used as scaffold material for periodontal regeneration [78]. In addition, mesenchymal stem cells derived from the dental papilla of the transplanted tooth periodontal tissues are applied to the apexes of the immature transplanted teeth [79].

Furthermore, regeneration materials such as platelet-rich plasma (PRP) used as scaffold materials [80] can be used inside the alveolus of the recipient site in order to favor the revascularization of the transplanted tooth as reported by Gaviño et al. (2020) [81] and also confirmed by a case series on 11 transplanted teeth by Gonzalez-Ocasio and Stevens [82].

### 4.5. Receiving Site

One of the prerequisites for a successful transplant is a sufficiently large recipient site [47,83]; the mesiodistal size of the tooth to be transplanted should be similar to that of the recipient area [11]. The socket is prepared to be slightly larger than the donor socket using round surgical burs at low speed and cooling with saline solution. By placing the tooth in the socket with slight pressure, the correspondence between the donor tooth and the recipient site is periodically checked. Obstacles in the socket wall are removed as soon as they are encountered [4].

There may be variations in the recipient site due to the time of tooth loss. In the case of transplantation in a post-extraction site of a freshly extracted tooth, a sufficient amount of bone is usually available. If the tooth was extracted a few months earlier, with partial or complete bone regeneration, an adequate recipient site can normally be created with burs [84,85]. In the presence of a marked atrophy of the alveolar process due to aplasia of a tooth, or its early loss, insufficient support would exist for the transplant. Additionally, when the donor tooth is positioned in the recipient site with an inadequate buccolingual space, root protrusion may occur through bone dehiscence and resorption of the alveolar ridge [86]. In all reported cases of tooth autotransplant failure, the recipient site was narrow; thus, the lack of buccal cortex and a narrow recipient site are considered risk factors for failure of the operation [30].

Although the results of one study indicate that the periodontal ligament of autotransplanted teeth has the potential to induce the formation of alveolar bone [68], the use of free bone autografts is recommended in cases of atrophy of the alveolar process [87]. Another approach to operating in the presence of an insufficient amount of bone is splitting osteotomy. In a study comparing splitting osteotomy with dental transplants using bone autografts and surgically created alveolar transplants, splitting osteotomy had a higher rate of inflammatory root resorption and a lower success rate [36].

A dimensional inconsistency between the recipient site and the root morphology of the transplanted tooth can lead to pulp necrosis [88]. Other authors also underlined the low success rate of teeth transplanted from the maxilla to the mandible [89], and in maxillary third molars that were transplanted in place of the maxillary first molars [90], probably due to the different morphology of the respective roots. The authors in [91] suggested a correlation between revascularization defects and increased distance between the root apex and the alveolar surface.

Adequate bone support with sufficient attached keratinized tissue is required at the recipient site to allow stabilization of the tooth [92]. The recipient site should be free from infection and/or inflammation. For this reason, although many protocols suggest that the removal of the problem tooth and the autotransplant should be performed in the same session [4], there are situations in which the autotransplant procedure should be postponed, as in the case of a periapical lesion [34]. In proximity to an anatomical structure such as the mandibular canal, the mental foramen, or the maxillary sinus, a periapical lesion could limit the curettage procedure [93]. In addition, residues of inflamed tissue could damage the repair processes of bones and soft tissues after autotransplantation; postponing the second phase of the autotransplant procedure from 8 to 12 weeks will probably result in a site free of inflammation and more comfortable handling of the recipient site due to the immaturity of the osteogenic bone [34].

### 4.6. Tooth Positioning and Stabilitation

The positioning of the donor tooth in the recipient site should establish a biological width similar to that of a naturally erupted tooth [94]. The donor tooth, once positioned in the recipient site, should be kept out of occlusal contact to prevent occlusal forces acting on the tooth from interfering with healing of the periodontium after transplantation [74].

Obtaining a suitable closure of the gingival flap around the transplanted tooth is a decisive procedure to ensure surgery is successful. The donor tooth reattachment depends significantly on the lack of bacterial invasion of the clot between the root and the alveolus and, in some cases, it is necessary to refine the flap and suture it before the tooth is inserted into the alveolus [95].

After placement, the tooth must be stabilized; the effect of the type of stabilization on periodontal healing remains controversial [96]. Various techniques for stabilizing transplanted teeth have been described, such as fixation with orthodontic brackets, ligatures, sutures, and composite resins [22,41], in which the period of immobilization varies from 1 [97,98] to 4–6 weeks [99,100].

It was originally thought that splinting could cause periodontal regeneration, with fixation periods of up to 3 months and using rigid splints [84,100]; however, it appears that rigid splinting of the transplanted tooth can lead to disturbances in pulp revascularization [49,101]. Some authors state that the formation of new vessels is stimulated by small movements during the function in the transplanted tooth and that the inhibition of the mobility of the transplant by a rigid splint exerts a negative influence on revascularization. This could explain the frequent episodes of pulp necrosis in rigidly splinted teeth [49,101]. It has also been reported that, due to the reduced vascularization of the transplanted tooth, there is a nutritional deficiency of the Hertwig epithelial sheath, thus influencing early or immature root development. This is likely because teeth stabilized with rigid splinting are positioned more superficially to avoid interference between the wire or composite and the gum; as a result, the greater distance between the base of the socket and the roots causes development problems in the Hertwig sheath [40].

Splints can also affect oral hygiene and periodontal regeneration, leading to complications, for example, inflammatory root resorption or ankylosis, with the risk of compromising the long-term results of the procedure [51,88].

In other studies, short-term flexible splinting appears to be more favorable, stabilizing the donor tooth with sutures for a period of 7–10 days [40,97]. Additionally, no stabilization has been used, with retention provided by the friction with adjacent teeth and with the alveolus prepared in such a manner to ensure maximum contact with the donor tooth [11,15].

A more rigid splint is appropriate if the donor tooth shows low stability [29,102]. The splint must be carefully selected because it plays a fundamental role in terms of the overall success of the procedure [50].

The occlusion should be checked to ensure that there is no occlusal interference; the occlusal adjustment should be more or less conservative, after which the type of restoration needed to adjust the occlusion and/or the aesthetic appearance of the tooth crown must be evaluated. An X-ray is taken before surgery, and before and after splinting, to assess the position of the donor tooth in the new site. Surgical dressing (periodontal packing) is applied to protect the graft against infection during the first 2–3 days of wound healing. This dressing is removed approximately 3–4 days after surgery [94].

Post-operative indications are required through instruction in oral hygiene and regarding diet, especially for the first post-operative week. Therefore, a follow-up is generally scheduled after 7–10 days for the removal of the sutures [39].

### 4.7. Endodontic Treatment

Following autotransplantation of teeth with complete root formation, endodontic treatment must be carried out to prevent infection of the pulp from spreading from the periapical area and the consequent inflammatory resorption of the root. This is necessary because revascularization of the pulp is not expected in teeth with fully formed roots [3,45,46].

A critical factor for apical inflammatory resorption following autotransplantation is infection of the root canal system. For this reason, closed apex tooth canals require pulp extirpation within 1–2 weeks of transplantation to avoid pulp infection followed by periapical inflammation and consequent inflammatory resorption of the root [47]. It has been reported that only 15% of teeth with a closed apex are revitalized following the autotransplant procedure, in contrast to 96% of teeth with an open apex [98]. The two-week period is chosen to minimize trauma to the periodontal ligament in the healing phase of the initial reattachment, whereas a longer period would increase the possibility of inflammatory resorption secondary to pulp infection [4,103].

If the donor tooth is easily accessible, it is possible to treat the root canal before its atraumatic extraction [4,104]. Third molars often present anomalies in the conformation of the roots and root canals, making endodontic treatment difficult [105,106,107]. In these cases, the apical resection of the tooth to be transplanted allows the removal of the more complex part of the root and limits the possible complications of a future orthograde treatment [29]. With the removal of the apical part of the root and the achievement of an apical seal by retrograde endodontic, better disinfection, cleansing, and shaping of the endodontic space is achieved due to the presence of straight, wide, and significantly shorter canals [43].

Intra-canal dressings during endodontic treatment performed with calcium hydroxide can promote the healing process and root resorption due to the high pH and antimicrobial properties [8,108,109]. In addition, in teeth with an immature apex, these dressings stimulate the deposition of mineral tissue with the consequent apical closure [42,110].

Extraoral endodontic treatment should be avoided because it can increase the time interval between extraction and transplantation. There is a risk of damaging the fibers and cells of the periodontal ligament when performing extraoral endodontic treatment [10]. A recent study conducted by Lin et al. 2020 on 1811 autotransplanted third molars found a higher survival rate for teeth whose endodontic treatment was performed after the transplant, compared to extra-oral or preoperative endodontic treatment [44].

In contrast, Boschini et al. 2020 presented opposing findings in a case report in which they stated that if adequate sterility is maintained during apicoectomy and retrograde closure of the canals, orthograde endodontic treatment can be avoided or delayed [42].

## 5. Conclusions

Third-molar autotransplantation is a valid solution to replace missing teeth. The key to this technique is the surgical procedure, which must be as atraumatic as possible to preserve the periodontal ligament of the tooth to be transplanted. The success rate is also linked to the stage of development of the root, with a worse prognosis in the case of a complete root. These cases require an endodontic treatment after about 2 weeks.

## Figures and Tables

**Figure 1 bioengineering-08-00120-f001:**
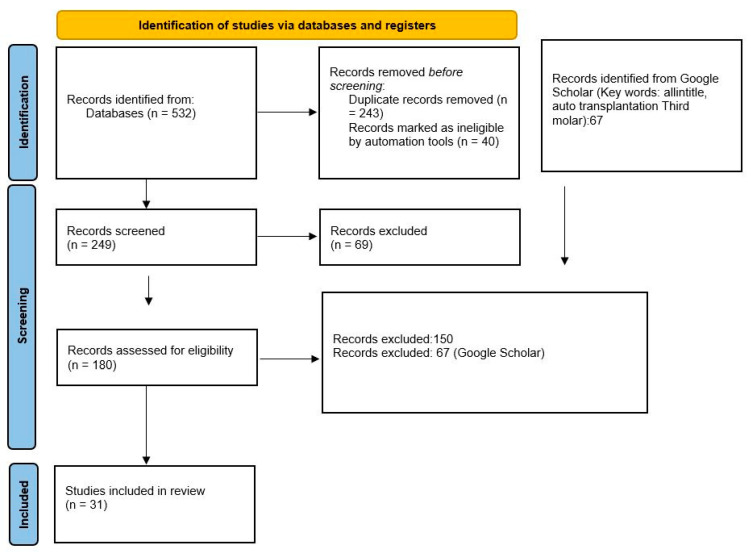
Flow chart of the different phases of the review.

**Figure 2 bioengineering-08-00120-f002:**
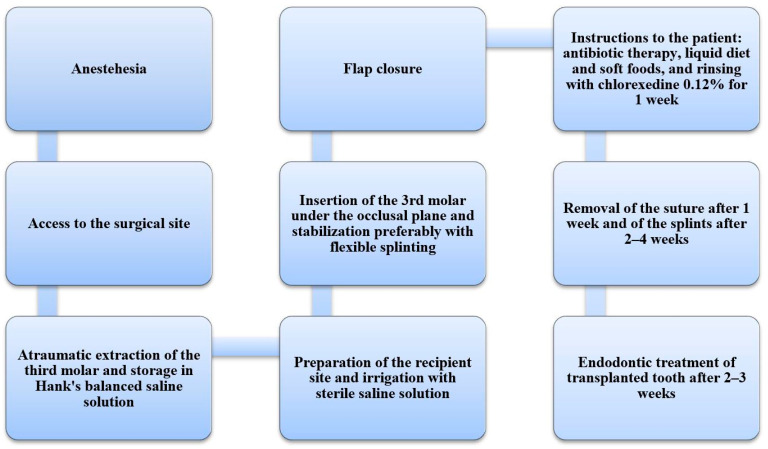
Flow chart of all surgical steps and procedures.

**Figure 3 bioengineering-08-00120-f003:**
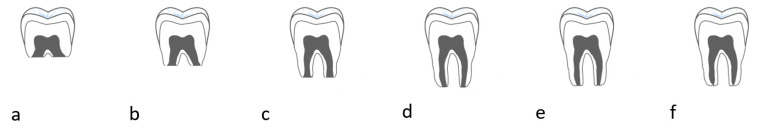
Stages of development of the root of a molar according to Moorrees CFA et al. (1963) [62]: (**a**). stage 1: roots developed to one-quarter of their length; (**b**). stage 2: roots developed to one-half of their length; (**c**). stage 3: roots developed to three-quarters of their length; (**d**). stage 4: roots developed along their entire length with open apices; (**e**). stage 5: root apices half closed by a wide periodontal ligament; (**f**). stage 6: roots with closed apices.

**Table 1 bioengineering-08-00120-t001:** Complete overview of the search methodology. Records identified by databases and Google Scholar: 599; Records after the application of the initial eligibility criteria: 249; Articles deemed potentially eligible: 180; Articles after the application of the inclusion and exclusion criteria: 78; Articles selected for qualitative analysis: 31.

Database Provider	Keywords	Search Details	Number of Records	Records after Removing Overlapping Articles
PubMed	third molar autotransplantation	Search number, Query, Sort By, Filters, Search Details, Results, Time: third molar autotransplantation, Most Recent, “(““molar, third”“[MeSH Terms] OR (““molar”“[All Fields] AND ““third”“[All Fields]) OR ““third molar”“[All Fields] OR (““third”“[All Fields] AND ““molar”“[All Fields])) AND (““autotransplantion”“[All Fields] OR ““transplantation, autologous”“[MeSH Terms] OR (““transplantation”“[All Fields] AND ““autologous”“[All Fields]) OR ““autologous transplantation”“[All Fields] OR ““autotransplantation”“[All Fields] OR ““autotransplantations”“[All Fields])”, 167, 02:46:48	167	167
PubMed	tooth autotransplantation	Search number, Query, Sort By, Filters, Search Details, Results, Time: “““tooth autotransplantation”““, Most Recent, “““tooth autotransplantation”“[All Fields]”, 90, 02:51:31	90	61
Scopus	third molar autotransplantation	TITLE-ABS-KEY (third AND molar AND autotransplantation)	165	43
Scopus	tooth autotransplantation	TITLE-ABS-KEY (“tooth autotransplantation”)	110	18
Google Scholar	auto transplantation Third molar	(Key words: allintitle, auto transplantation Third molar)	67 ^1^	0
Total	\	\	599	289
Articles excluded	\	\	0	310

^1^ The Google Scholar records were subsequently evaluated and manually excluded due to the impossibility of implementing all the records in EndNote. \ = Number not available.

## Data Availability

Not applicable.

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
