# Peer review of "Autotransplantation of the Third Molar: A Therapeutic Alternative to the Rehabilitation of a Missing Tooth: A Scoping Review"

_bioengineering, 2021, doi:10.3390/bioengineering8090120_

Round 1
Reviewer 1 Report
The aim of this scoping review was to provide the most up-to-date information and data on the clinical principles of the third-molar autograft and thus provide clinical considerations for its success.
Third-molar auto transplantation is a valid solution to replace missing teeth.
Overall the current review is well written and the topic is interesting, but still some small changes are needed.
The quality of the Figures need to be improved, is impossible to see the images because is out of focus.
Regarding the Figure 1 and 2. the quality need to be improved, 300 dpi.
the author focus their review on the auto transplantation as valid solution for replacing missing teeth.
in the line 231: "third molars have a rich supply of blood and stem cells,
root development following transplantation depends on the preserved activity of Hertwig's epithelial sheath". Based on this, it would be nice to add a paragraph of the potential of oral stem cells and their regeneration capacity "Oral bone tissue regeneration: Mesenchymal stem cells, secretome, and biomaterials" published by Gugliandolo et al will be definetely hellfull for this purpose.
line 301: The periodontal ligament has cells that genetically have the ability to differentiate into fibroblasts, osteoblasts, and cementoblasts. It is strongly suggest to read the paper published by Trubiani et al. "Human oral stem cells, biomaterials and extracellular vesicles: A promising tool in bone tissue repair" , " Functional relationship between osteogenesis and angiogenesis in tissue regeneration " published by Diomede F.et al , and "Ascorbic Acid: A New Player of Epigenetic Regulation in LPS- gingivalis Treated Human Periodontal Ligament Stem Cells " published by Marconi et al., to add a sentence on human periodontal stem cells, their features and their crucial role in tissue engineering.
Author Response
Reviewer 1
The aim of this scoping review was to provide the most up-to-date information and data on the clinical principles of the third-molar autograft and thus provide clinical considerations for its success.
Third-molar auto transplantation is a valid solution to replace missing teeth.
Overall the current review is well written and the topic is interesting, but still some small changes are needed.
The quality of the Figures need to be improved, is impossible to see the images because is out of focus.
Regarding the Figure 1 and 2. the quality need to be improved, 300 dpi.
the author focus their review on the auto transplantation as valid solution for replacing missing teeth.
in the line 231: "third molars have a rich supply of blood and stem cells,
root development following transplantation depends on the preserved activity of Hertwig's epithelial sheath". Based on this, it would be nice to add a paragraph of the potential of oral stem cells and their regeneration capacity "Oral bone tissue regeneration: Mesenchymal stem cells, secretome, and biomaterials" published by Gugliandolo et al will be definetely hellfull for this purpose.
line 301: The periodontal ligament has cells that genetically have the ability to differentiate into fibroblasts, osteoblasts, and cementoblasts. It is strongly suggest to read the paper published by Trubiani et al. "Human oral stem cells, biomaterials and extracellular vesicles: A promising tool in bone tissue repair" , " Functional relationship between osteogenesis and angiogenesis in tissue regeneration " published by Diomede F.et al , and "Ascorbic Acid: A New Player of Epigenetic Regulation in LPS- gingivalis Treated Human Periodontal Ligament Stem Cells " published by Marconi et al., to add a sentence on human periodontal stem cells, their features and their crucial role in tissue engineering.
Answer
Thank you for reviewing the manuscript and for your comments and suggestions
- The dpi of figures 1 and 2 have been increased to 400 dpi
- the following paragraph has been added as suggested :In this regard, a review of the literature conducted by Gugliandolo et al highlighted the possible use in tissue bioengineering of oral mesenchymal stem cells in particular deriving from dental tissues, such as dental pulp stem cells, stem cells from the apical papilla (, periodontal ligament stem cells , gingival-derived steam cells, dental follicle stem cells, tooth germ stem cells) [1].
The following paragraph has been added as suggested: In particular, stem cells that are collected by scraping the alveolar ridge and the horizontal fibers of the ligamentous periodontal tissue of dental elements with healthy tissue play a fundamental role in the tissue regeneration of the periodontal ligament as reported in a recent review of the literature conducted by Trubiani et al [2,3] role partially confirmed by a subsequent study by Marconi et al. [4]
- Gugliandolo, A.; Fonticoli, L.; Trubiani, O.; Rajan, T.S.; Marconi, G.D.; Bramanti, P.; Mazzon, E.; Pizzicannella, J.; Diomede, F. Oral Bone Tissue Regeneration: Mesenchymal Stem Cells, Secretome, and Biomaterials. Int J Mol Sci 2021, 22, doi:10.3390/ijms22105236.
- Trubiani, O.; Marconi, G.D.; Pierdomenico, S.D.; Piattelli, A.; Diomede, F.; Pizzicannella, J. Human Oral Stem Cells, Biomaterials and Extracellular Vesicles: A Promising Tool in Bone Tissue Repair. Int J Mol Sci 2019, 20, doi:10.3390/ijms20204987.
- Diomede, F.; Marconi, G.D.; Fonticoli, L.; Pizzicanella, J.; Merciaro, I.; Bramanti, P.; Mazzon, E.; Trubiani, O. Functional Relationship between Osteogenesis and Angiogenesis in Tissue Regeneration. Int J Mol Sci 2020, 21, doi:10.3390/ijms21093242.
- Marconi, G.D.; Fonticoli, L.; Guarnieri, S.; Cavalcanti, M.; Franchi, S.; Gatta, V.; Trubiani, O.; Pizzicannella, J.; Diomede, F. Ascorbic Acid: A New Player of Epigenetic Regulation in LPS-gingivalis Treated Human Periodontal Ligament Stem Cells. Oxid Med Cell Longev 2021, 2021, 6679708, doi:10.1155/2021/6679708.

Reviewer 2 Report
The authors performed a scoping review on "autotransplantation of the third molar". However, the authors did not describe the differences between systematic and scoping review. Line 104: The protocol and methodology adopted in this review have already been used in other systematic reviews conducted by the same research group. Could the authors explain how their use of scoping review was different from systematic review?
This manuscript contains many clinical techniques oriented about "how to" perform autotransplantation.
Many parts of the manuscript are difficult to understand. For example, line 471: The portion of the root that presents the most critical issues is the apical one-third [102]. The authors need to explain this statement.
Furthermore, the items listed below are not clear.
Figure 1. Flow chart is unreadable.
Figure 2. Flow chart of all surgical steps and procedures is unreadable.
Figure 3. Stages of development of the root of a molar according to Moorrees CFA et al (1963); (a) stage 1: roots developed to one quarter of their length; (b) stage 2: roots developed to one half of their length; (c) stage 3: roots developed to three-quarters of their length;....
Comments:
Moorrees CFA et al (1963) should be cited as reference number. The diagram needs to be improved to reflect 1/4, 1/2 and 3/4 of their length. Moorrees's original diagram was better to display the relative length.
Author Response
Reviewer 2
The authors performed a scoping review on "autotransplantation of the third molar". However, the authors did not describe the differences between systematic and scoping review. Line 104: The protocol and methodology adopted in this review have already been used in other systematic reviews conducted by the same research group. Could the authors explain how their use of scoping review was different from systematic review?
This manuscript contains many clinical techniques oriented about "how to" perform autotransplantation.
Many parts of the manuscript are difficult to understand. For example, line 471: The portion of the root that presents the most critical issues is the apical one-third [102]. The authors need to explain this statement.
Furthermore, the items listed below are not clear.
Figure 1. Flow chart is unreadable.
Figure 2. Flow chart of all surgical steps and procedures is unreadable.
Figure 3. Stages of development of the root of a molar according to Moorrees CFA et al (1963); (a) stage 1: roots developed to one quarter of their length; (b) stage 2: roots developed to one half of their length; (c) stage 3: roots developed to three-quarters of their length;....
Comments:
Moorrees CFA et al (1963) should be cited as reference number. The diagram needs to be improved to reflect 1/4, 1/2 and 3/4 of their length. Moorrees's original diagram was better to display the relative length.
Answer
Thank you for reviewing the manuscript, your advice has been very helpful in improving the manuscript.
- The text has been changed and the following sentence has been deleted: Line 104: The protocol and methodology adopted in this review have already been used in other systematic reviews conducted by the same research group. The following sentence was added by modifying the previous one This review was conducted based on PRISMA-ScR (Preferred Reporting Items for Systematic reviews and Meta-Analyses extension for Scoping Reviews)[1]
- The dpi of figures 1 and 2 have been increased to 400 dpi
- Figure 3 has been redrawn according to the reviewer's suggestions. Furthermore, the bibliographic reference has been added as suggested [2]
- Tricco, A.C.; Lillie, E.; Zarin, W.; O'Brien, K.K.; Colquhoun, H.; Levac, D.; Moher, D.; Peters, M.D.J.; Horsley, T.; Weeks, L., et al. PRISMA Extension for Scoping Reviews (PRISMA-ScR): Checklist and Explanation. Ann Intern Med 2018, 169, 467-473, doi:10.7326/m18-0850.
- Moorrees, C.F.; Fanning, E.A.; Hunt, E.E., Jr. FORMATION AND RESORPTION OF THREE DECIDUOUS TEETH IN CHILDREN. Am J Phys Anthropol 1963, 21, 205-213, doi:10.1002/ajpa.1330210212.
